# Numerical simulations suggest asteroids (101955) Bennu and (162173) Ryugu are likely second or later generation rubble piles

K. J. Walsh [1] ✉, R-L. Ballouz [2], W. F. Bottke [1], C. Avdellidou[3,4], H. C. Connolly Jr [5,6,7], M. Delbo[3], D. N. DellaGiustina [6], E. R. Jawin[8], T. McCoy[9], P. Michel [3,10], T. Morota [10], M. C. Nolan [6], S. R. Schwartz [11], S. Sugita [10] & D. S. Lauretta [6]

Rubble pile asteroids are widely understood to be composed of reaccumulated debris following a catastrophic collision between asteroids in the main asteroid belt, where each disruption can make a family of new asteroids. Near-Earth asteroids Ryugu and Bennu have been linked to collisional families in the main asteroid belt, but surface age analyses of each asteroid suggest these bodies are substantially younger than their putative families. Here we show, through a coupled collisional and dynamical evolution of members of these families, that neither asteroid was likely to have been created at the same time as the original family breakups, but rather are likely remnants of later disruptions of original family members, making them second, or later, generation remnants. Our model finds about 80% and 60% of asteroids currently being delivered to near-Earth orbits from the respective families of New Polana and Eulalia are second or later generation. These asteroids delivered today in the 0.5-1 km size range have median ages since their last disruption that are substantially younger than the family age, reconciling their measured crater retention ages with membership in these families.

Near-Earth asteroids (NEAs) are dynamically transient bodies that primarily originate in the main asteroid belt and survive for only an average of 10 Myr on near-Earth orbits before either impacting the Sun or a planet or being ejected from the solar system[1]. They transition from the Main Belt to near-Earth orbits through a handful of dynamical pathways that are typically orbital resonances with giant planets or Mars[2]. While any given NEA orbit is chaotic and impossible to directly trace back to a specific orbit or location in the Main Belt, it is possible, in some cases, to determine which dynamical pathway an NEA most likely followed[2,3].

NEAs larger than a few hundred meters and smaller than 5–10 km are expected to be rubble piles, which are the reaccumulated debris from the collisional fragmentation of larger asteroids[4–6]. Models of the collisional history of the asteroid belt have shown that bodies smaller than diameter $D \sim 10$–$20$ km have collisional lifetimes shorter than the age of the solar system[7]. They are unlikely to have survived in their current state and are vastly more likely to be remnants of a more recent collisional event.

Bennu and Ryugu, rubble pile NEAs that have both been visited by spacecraft[8,9], are highly likely to reach their specific orbits in near-Earth

[1]Southwest Research Institute, Boulder, CO, USA. [2]Applied Physics Laboratory, Johns Hopkins University, Laurel, MD, USA. [3]Laboratoire Lagrange, Université Côte d'Azur, Observatoire de la Côte d'Azur, CNRS, Laboratoire Lagrange, Nice, France. [4]School of Physics and Astronomy, University of Leicester, Leicester LE1 7RH, UK. [5]Dept. of Geology, Rowan University, Glassboro, NJ, USA. [6]Lunar and Planetary Laboratory, University of Arizona, Tucson, AZ, USA. [7]American Museum of Natural History, New York, NY, USA. [8]Smithsonian Institution National Air and Space Museum, Washington, DC, USA. [9]Smithsonian Institution National Museum of Natural History, Washington, DC, USA. [10]University of Tokyo, Tokyo, Japan. [11]Planetary Science Institute, Tucson, AZ, USA. ✉e-mail: kwalsh@boulder.swri.edu

space via the $v_6$ secular resonance at the inner edge of the Asteroid Belt or via the Intermediate Mars Crossing population (IMC). The IMC population is filled largely by minor resonances, such as the overlapping resonances with Jupiter and Mars at 2.25au (hereafter J7:2M5:9). Models of NEA delivery find that the probability of a body on Bennu's current orbit has 82% and 18% probability to originate from the $v_6$ or IMC, respectively[10]. Ryugu has similar origins with 80% and 20% likelihood from the same pathways[11]. Updated NEO models, using a larger number of dynamical sources find similarly high probabilities of delivery from $v_6$ resonance where Bennu is 86% likely and Ryugu 88% (refs. [12,13]).

Evidence of on-going asteroidal collisional evolution can be found by way of asteroid families, which litter the Main Belt (see refs. [14,15] for reviews). Here, the reaccumulated remnants of disruption events are linked by their similar orbital properties and sometimes through their physical properties such as spectral colors and albedo. Therefore, nearly all small ($D < 10$ km) asteroids in the Main Belt should be remnants of these big collisions, share a genetic link with a large parent body, and have a potentially huge number of relatives (family members) formed at the same time in the same event. Once a family is formed all the new members are subject to the same collisional environment as the parent body. The clock on the collisional lifetime of each new member starts ticking and eventually many will experience big impacts, resulting in the formation of new sub-families within the larger original family. It is common to find numerous smaller sub-families within larger, older, families (e.g refs. [16,17]), and there are certainly large numbers of families and sub-families that are yet to be detected.

Similarly, all the new family members made in a collision are subject to size-dependent orbital drift due to the Yarkovsky effect[18,19]. The rate of semi-major axis change depends on an asteroid's obliquity, thermal properties, and distance from the Sun, but importantly has a $1/D$ dependence, where $D$ is the diameter of the asteroid, that drives the smallest members of a family to drift farther and faster than larger members. Constraints on drift rates allow for the spread of a family to act as a chronometer and allow estimates of families' ages, or time of formation (see refs. [19–24]) This semi-major axis drift is also the mechanism by which Main Belt asteroids transition to near-Earth orbits, as their drift often passes through orbital resonances with the giant planets or Mars. This can cause their orbits to be dynamically excited, i.e. their orbital eccentricity is increased, and some of them may subsequently encounter a terrestrial planet and be scattered into near-Earth space.

The knowledge of the dynamical pathway, combined with other spectroscopic and radiometric constraints, led to links between Bennu and Ryugu with the New Polana or Eulalia families[10,11,25,26]. These low-albedo families are found in the outer edge of the inner Main Belt (2.1–2.5 au), overlap with each other in their distribution of orbital elements and appear nearly indistinguishable in visible and near-infrared spectroscopic studies[27–30]. It is estimated that New Polana formed 1400±150 My ago and Eulalia formed $830^{+370}_{-100}$ Myr ago[26], and either is currently capable of delivering 0.5–1 km objects like Bennu and Ryugu to the near-Earth population through the $v_6$ or IMC pathways.

An asteroid that formed after the initial collision that led to the formation of these families must endure throughout the entire lifespan of the family without being shattered by subsequent collisions or dynamically expelled before reaching its current near-Earth orbit. In the collisional environment of the Main Belt, asteroids the size of Bennu ($D \sim 500$ m) or Ryugu ($D \sim 1$ km) are estimated to have lifetimes against collisional disruption of only ~200 Myr and 440 Myr, respectively[7,31]. To be 1st-generation members of either family they must survive several collisional lifetimes. The alternative is that a larger 1st-generation remnant that was more resilient to collisional disruption due to its larger size, but that drifted more slowly for many millions of

years, was itself eventually disrupted and produced Bennu or Ryugu as part of a second, or later, generation sub-family.

This difference has implications for interpreting the observed geology on both bodies and interpreting the nature of the returned samples. The crater retention age for Bennu and Ryugu are 10-65 Myr (ref. [32]) and <30 Myr, respectively[8,33]. If it is probable that they are second generation or older remnants of a family then their young surfaces may be reconciled with their proposed membership in these two asteroid families. If not, then membership in these families is challenged or geologic processes such as global resurfacing event(s) would need to be responsible for resetting their crater retention ages. Meanwhile, indications of a few or numerous large collisional events in the returned samples could illuminate a likely history or reveal gaps in modeling assumptions.

In this work we present results from a combined dynamical and collisional model of asteroid family evolution. It shows that it is more likely that asteroids currently being delivered from the Eulalia and New Polana families are second, or later, generation remnants and not 1st generation remnants.

## Results
### Modeling family generation and evolution
Previous efforts have carried out extensive N-body simulations of members of the New Polana and Eulalia families drifting across the main asteroid belt[26]. Those N-body simulations included the planets and thus incorporated the effects of the orbital resonances directly, but did not account for collisional removal or the generation of new simulated bodies in collisions. Other models have developed sophisticated routines to capture the effects of collisional evolution, and its large-scale effects on the size frequency distribution (SFD) of the asteroids in the main asteroid belt[7,34]. These large models cover such large timescales and large swaths of the Main Belt that they don't take into account individual asteroid families and do not incorporate the ability to track individual asteroids and their changing orbits. The modeling approach here, like the former, tracks individual asteroids' orbital drift and their removal by resonances, and like the latter, also considers their collisional disruption and the re-distribution of mass to smaller bodies.

Asteroid families are modeled here through Yarkovsky effect-driven orbit drift and a Monte-Carlo approach to collisional evolution. In this model, the initial asteroid family, and those created through later disruption, have half of the disrupted target's mass in its largest remnant[7]. The next largest body, the largest fragment is modeled to always be a constant mass ratio of the largest remnant[7]. It is the largest fragment that is the anchor for the power law size-distribution of asteroids for the rest of the family (see Fig. 1). Previous models of the Main Belt SFD evolution used the largest size ratio of 0.8 (where mass of the largest fragment is ½ that of the largest remnant) and re-created many aspects of the Main Belt size distribution[7], while others left it as a free parameter that varied as a function of the impact properties[34]. Here, the largest fragment relative to the largest remnant is left as a free parameter, varying between 0.2, 0.4, 0.6, and 0.8 the size of the largest remnant (0.008, 0.064, 0.022 and 0.5, respectively, the mass). This range includes the previously used values and brackets the observed values for a handful of well-studied inner Main Belt C-type families and is in line with numerical simulations of disruption and reaccumulation[4,35,36] (Fig. 1). The models here assume that the slope of the cumulative SFD is constant, similar to ref. [34], and unlike ref. [7] who used a piecewise SFD. Values for that slope found in simulations[36] and in observed asteroid families span a wide range[14]. Again, the power law exponent values for family SFD slope were treated as a free variable. Slope values of −3, −4, and −5 were tested to bracket the most likely typical initial family SFDs.

To initiate a simulation the diameter of the original parent body is selected and the fragment sizes are then determined using the formula

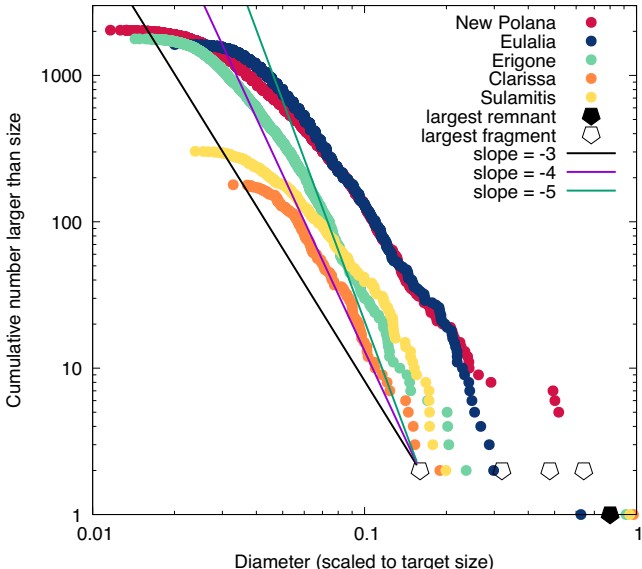

**Fig. 1 | Observed asteroid family size-frequency distributions.** Observed inner Main Belt C-type asteroid families of various size and age are shown with their cumulative number as a function of a scaled diameter, taken to be the sizes of single objects that correspond to each family's integrated mass[25,75]. The families and their ages are Erigone (green circles) at 130±30 Myr, Clarissa (orange circles) at roughly 60±10 Myr, Sulamitis (yellow circles) at 200±40 Myr, Eulalia (blue circle) at 830$^{+370}_{-100}$ Myr and New Polana (red circles) at 1400±150 Myr (ref. 26). The rollover of the SFD at small member sizes is likely an artifact due to completeness limits in the asteroid catalogs. The large, closed, symbol is the modeled value for the size of the largest remnant (where it is ½ the total mass and thus it is 0.8 the scaled diameter) and open large symbols the range of largest fragment sizes for comparison (which correspond to cumulative number 2 on the y-axis). Finally, the three SFD slopes used in the modeling are shown, where they are shown originated from one of the open symbols.

for largest remnant, largest fragment and the power law slope for the remaining distribution of family members. Nominal settings utilize a largest remnant to be 0.8 times the size of the target (1/2 the mass) and the largest fragment to be 0.4 times the size of the largest remnant and a power-law SFD slope of −4.0. The size of the parent body is not precisely known for either family but is constrained by the size of the observed largest family members, (495) Eulalia and (142) Polana (with diameters of about 40 km and 55 km respectively), and by the nature of their family SFDs showing that neither family appeared to be super-catastrophic in nature[25]. Integrating the total mass of known and possible family members and accounting for various loss mechanisms, the estimated parent body for the Eulalia family had a diameter $D_{pb}$ = 100-160 km (ref. 25). The New Polana family is more challenging to estimate[25]. Given the asteroids similar size and their families' similar SFDs, here we consider simulating $D_{pb}$ = 80 km, 100 km and 160 km for both families to attempt to bracket all possibilities.

The collisional cascade in the Main Belt creates debris down to very small sizes, but Bennu has a diameter about 0.5 km and Ryugu about 1 km (refs. 37,38). Hence for the focus of this work it is only necessary to track the delivery of simulated asteroids down to those sizes, and not below. Collisional timescales for simulated asteroids are a function of the Main Belt background asteroid population as previously calculated[2], rather than the number, size, or orbits of the asteroids simulated here. Thus, ignoring very small objects does not change the collision calculation results.

The initial semi-major axes of the objects in a family are set to the same value as the parent bodies, where Eulalia has semi-major axis $a$ = 2.487 au and Polana $a$ = 2.41 au. There is some initial size-dependent dispersal of orbital semi-major axis that scales with the

escape velocity of the target bodies and this is included for each initial family and subsequent sub-family following the formulations of ref. 20. The objects are given random obliquities on a unit sphere that determine their Yarkovsky drift rate and direction when combined with their sizes. The evolution of the objects due to thermal torques is complex and poorly constrained at small sizes where spin rate and obliquity changes are expected due to the YORP effect[18,19,26]. Changes in obliquity, in turn, affect orbital drift rates. Thus, the asteroids were modeled in two ways. First, the initial obliquities did not change throughout the simulation and drift rates for each simulated asteroid were calculated from Bennu's measured rate of −284 m/year that is scaled to its diameter, semi-major axis, and obliquity[39,40]. Second, they were modeled with the YORP cycles and stochastic YORP functions and parameters described in ref. 26. These formulations require tracking spin rates, which have direct interplay with changing obliquity and reorientation due to impacts. The net effect is for obliquities to evolve towards 0 or 180°, with some reorientations due to impacts on very slowly spinning bodies[26]. Unless otherwise noted results are shown for the first modeling method with unchanging obliquities.

An asteroid in the main asteroid belt will experience collisions with a size distribution of impactors, and eventually could be the impactor itself when hitting a larger body. For this effort we do not attempt to accumulate every impact, rather we follow previous works that modeled main belt collisional evolution[26] and determine when a disruptive impact is likely to happen based on the collisional probabilities for the main asteroid belt. This is calculated via the size-dependent specific impact energy equal to $Q^*_d$, which is the ratio of the projectile's kinetic energy to the target's mass leading to half of the target's mass being in the largest remnant. The timescale for a disruption to occur is treated with a disruption law based on measured physical properties of boulders on Bennu[31] and is similar in these size ranges to that used in previous models of Main Belt collisional evolution[7,41].

When a simulated asteroid is disrupted it is replaced with a size distribution of smaller remnants. These small remnants follow the same rules for the largest remnant, largest fragment and SFD slope as for the initial family and explore the same range of values for each variable. The total integrated mass is limited so that no mass beyond that of the disrupted body is added to the simulation. Similarly, no bodies smaller than 0.5 km are added. Newly formed bodies are started near the location of their recently disrupted parent, whose size sets the scaling for the the size-dependent spread in the semi-major axis relative to the parent's. They are given random obliquities to determine their drift rate and direction.

Simulated asteroids can be removed via collisional evolution or by orbital dynamics—specifically if their orbits drift across resonances with planets. As a simulated asteroid drifts across the main belt, it can encounter mean motion or secular resonances with planets, and face certain, or probable removal from the asteroid belt by eccentricity increase. Drifting inward from the inner main belt, the $\nu_6$ secular resonance will be 100% efficient at removing a body if it reaches $a$ of ~2.15 au. The location of $\nu_6$ is inclination-dependent, and is, in places, close to the location where a body could start to cross the orbit of Mars, which is eccentricity-dependent. The Eulalia and New Polana families have relatively low eccentricity and inclination, allowing for a simple calculation of semi-major axis overlap with the location of $\nu_6$. The outer edge of the inner Main Belt is the 3:1 resonance with Jupiter, which, like $\nu_6$, is nearly 100% efficient at removing bodies from the belt. In the middle of the inner belt lies the overlapping Jupiter 7:2 Mars 5:9 mean motion resonances at $a$ = 2.255au (hereafter J7:2M5:9), which could be encountered from either direction (inward or outward drifting—depending on the simulated asteroid's starting location) and is about 33% efficient at removal of km-sized asteroids[26,42]. Note that the removal efficiency is drift rate-dependent, which itself is size dependent[26,43] and these removal calculations are valid only for the km-scale objects.

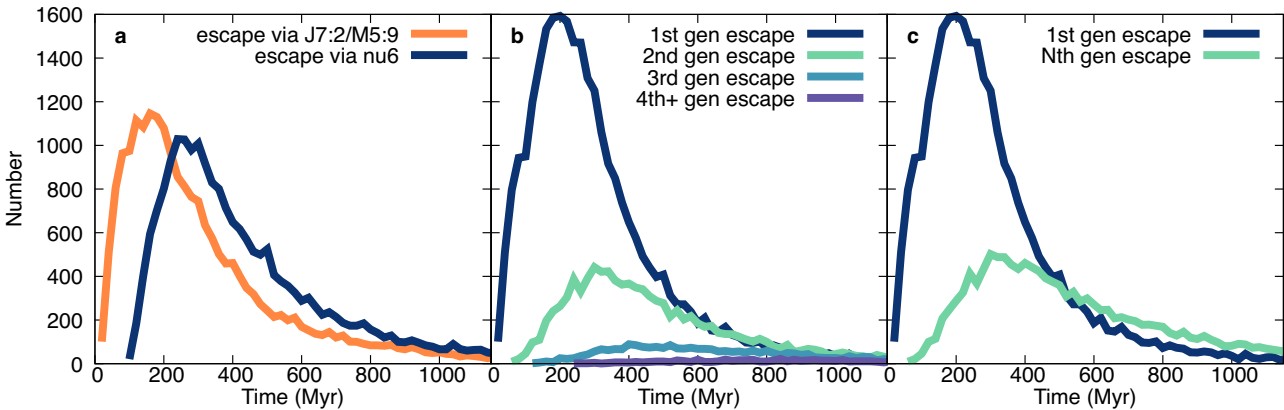

**Fig. 2 | Delivery pathways and generations over time for New Polana.** Delivery of bodies from the New Polana family with family SFD = −4 and second largest fragment diameter = 0.4 diameter of largest remnant and a 100 km parent body. **a** shows the relative number of simulated asteroids delivered by the J7:2M5:9 in orange and $\nu_6$ resonance in navy over time. **b** shows the delivery of 1st, 2nd, 3rd and 4th generation fragments over time in navy, green, blue and purple respectively and (**c**) shows the same with all later generation fragments as $N$th generation in green. The crossover time for this simulation is at 450 Myr, visible in (**c**) where $N$th generation line crosses the 1st generation line. $T$ = 0 Myr is the time of family formation.

Furthermore, while the $\nu_6$ and 3:1 are effectively 100% efficient at removing asteroids from the asteroid belt, neither are as efficient at placing them onto near-Earth orbits, as some are ejected from the Solar System or achieve some other fate. Here, since previous modeling has found Bennu and Ryugu's likely dynamical pathway to their current orbits, there is no need to blend the relative probability of NEA delivery between these two removal sources. Instead, we are exploring the collisional history of objects that follow this same dynamical path. Since this effort will rely on a Monte Carlo approach the total number of initial simulated asteroids in a family can be as large as computationally feasible as long as it adheres to the family size distribution. The numbers and sizes of fragments made in each subsequent collision and disruption must follow the same rules and make proportionally similar numbers, while still adhering to the family size distribution. Thus, nominal simulations below begin with 300,000 simulated asteroids, although the number grows throughout each simulation depending on the number and nature of the collisions in each.

### 1st- or Nth-generation remnants

The delivery of different generations of asteroidal fragments varies over time due to the different family locations relative to the important escape routes. For example, families farther from the J7:2M5:9 and $\nu_6$ take longer to deliver their first fragments, allowing more time for collisional evolution. The evolution for the New Polana family illustrates the time-varying delivery from the different pathways, where the family location at 2.41au requires asteroids to cross the J7:2M5:9 prior to reaching the $\nu_6$ resonance. This results in an initial spike at about 180 Myr that precedes a spike from $\nu_6$ coming nearly 100 Myr later (Fig. 2a). While the J7:2M5:9 is only 1/3 efficient at removing asteroids, the extra roughly 100 Myr that it takes them to reach the $\nu_6$ allows for many to be removed via further collisional evolution. As the time since family formation increases (family formation is at 0 Myr in the figures), the effects of collisional evolution increase the number of 2nd- and later-generation of fragments reaching the resonances (Fig. 2b, c).

A simple property extracted from each set of simulation parameters is the crossover time when the number of $N$th generation asteroids begin to dominate in number over 1st generation bodies. For the set of nominal parameters where the largest fragment is 0.4 the size (and 6% the mass) of the largest remnant and the family cumulative SFD slope is −4.0, this occurs at 450 Myr for the New Polana family with an assumed parent body size of 100 km (Fig. 2b,c). For the same parameters, the Eulalia family reaches the crossover time at 510 Myr,

due to its center location at 2.448 au, which is farther from either of the important resonances (Fig. 3b,c).

Delivery from the Eulalia and New Polana families have similar dependencies on the two parameters: where crossover times increase for 1) steeper family SFD slopes and 2) smaller largest fragment sizes (Fig. 4). The steeper family SFD slopes decrease the number of smaller bodies generated during each disruption event making the re-filling of the small end of the size distribution slower, which in turn decreases the number of fast-drifting bodies that can reach escape resonances. Similarly, the size of the largest fragment determines the mass of disrupted objects that are preferentially retained, but also can result in a large number of fragments being created below the resolution of the model at 0.5 km, where they are no longer tracked. The smaller largest fragments result in even more mass being put into asteroids with sizes below that limit and decrease the number of new, km-sized, fragments to be tracked, and subsequently causing later crossover times.

The New Polana family is estimated to be 1400±150 Myr (ref. 26) and there are many millions of years that 1st- generation fragments dominate delivery. The relevant delivery time is 1400 Myr after family formation and corresponds to asteroids escaping the Main Belt today. For the nominal modeling scenario, the delivery at 1400 Myr after family formation is 21% from 1st generation, compared to 41% 2nd generation and 22% 3rd generation (Table 1). The younger Eulalia family had a delivery composition of 38% 1st generation, 39% 2nd generation and 18% 3rd generation after 830 Myr (Table 1).

Across all tested ranges in the largest fragment size, family SFD slope and parent body size there was variation on order of a few 100 Myr for the crossover times (Fig. 4). The New Polana family has crossover times that range from 340 Myr to 520 Myr while Eulalia spans crossover times from 410 Myr to 610 Myr for the same suite of parameters. The inclusion of YORP cycles and stochastic YORP shortens the range of crossover times to 330 Myr to 430 Myr for New Polana and 400 to 540 Myr for Eulalia. YORP cycles lead to obliquities of 0 or 180°, maximizing drift inward or outward which result in shorter crossover times. The range of crossover times for both families, for all tested parameters with or without YORP cycles and stochastic YORP indicates that $N$th-generation objects dominate delivery to the inner solar system at timescales shorter than the expected family age.

### Formation ages of delivered asteroids

In the models presented here, the formation age of each simulated asteroid is the time when it enters the simulation from either the family-forming event or the subsequent disruption of a larger fragment. For

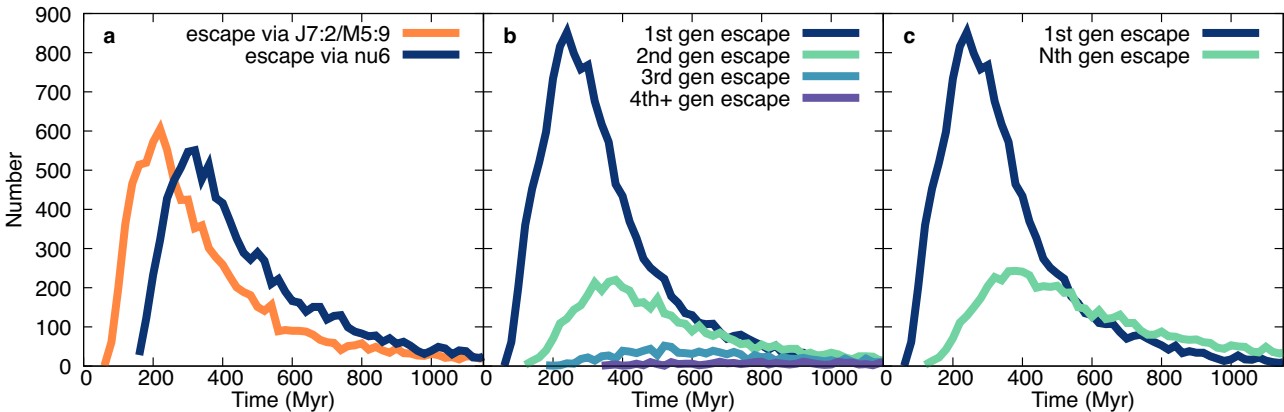

**Fig. 3 | Delivery pathways and generations over time for Eulalia.** Delivery of bodies from the Eulalia family with family SFD = −4 and second largest fragment diameter = 0.4 diameter of largest remnant and a 100 km parent body. **a** shows the relative number of simulated asteroids delivered by the J7:2M5:9 in orange and the ν6 resonance in navy over time. **b** shows the delivery of 1st, 2nd, 3rd and 4th generation fragments over time in navy, green, blue and purple respectively and (**c**) shows the same with all later generation fragments as Nth generation in green. The crossover time for this simulation is at 510 Myr, visible in (**c**) where Nth generation line crosses the 1st generation line. T = 0 Myr is the time of family formation.

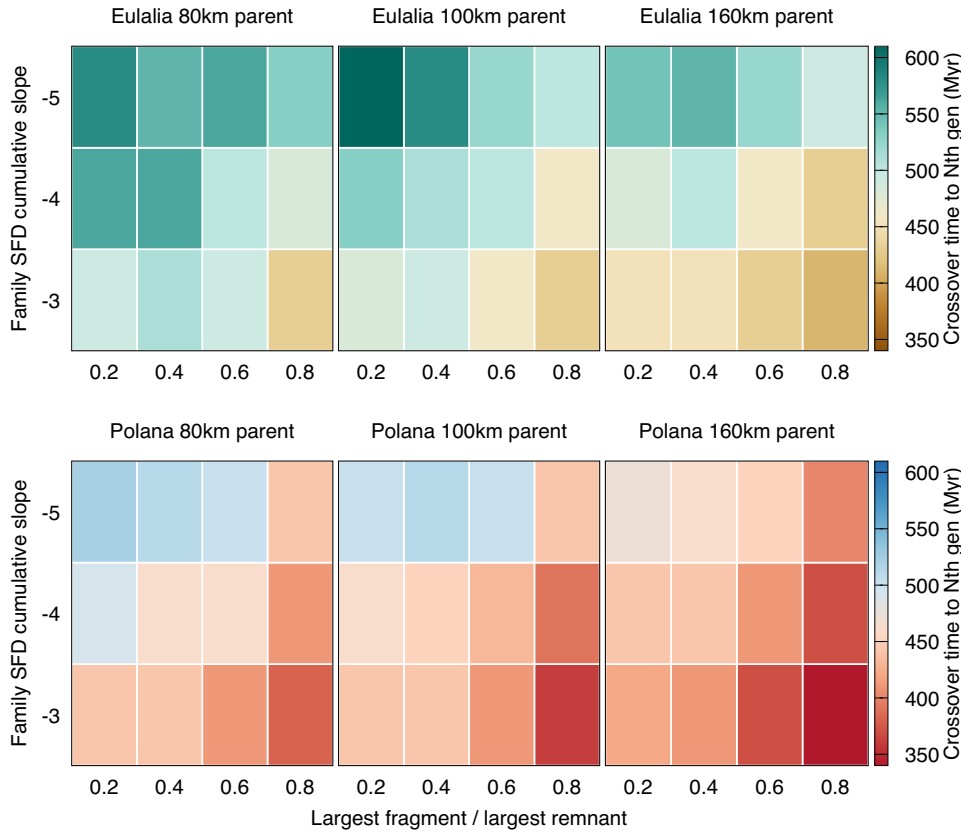

**Fig. 4 | Crossover times for New Polana and Eulalia families.** The measured crossover times for Nth generation fragments to dominate delivery for the New Polana and Eulalia families. The family SFD power law slope was varied from −3 to −5 and the largest fragment size ratio relative to the largest remnant was tested at 0.2, 0.4, 0.6 and 0.8. Parameters were then tested for parent bodies with diameter 80 km, 100 km and 160 km. The color for each simulation outcome corresponds to the crossover time when Nth generation fragments are dominating delivery over 1st generation.

the D < 1 km objects that reached the inward resonances in the nominal New Polana simulation 33% and 21% had asteroid formation ages younger than 60 Myr and 10 Myr, respectively, with a median age of 190 Myr (Fig. 5). For the Eulalia simulation 19% and 15% were younger than 60 Myr and 10 Myr respectively with a median asteroid age of 330 Myr. The differences primarily reflect the different family ages, where the New Polana family age of 1400 Myr is nearly three collisional

timescales for a 1 km object which greatly reduces the number of 1st generation, and very old objects, that could survive. The greater fraction of Nth generation fragments from New Polana (80%) decreases the median asteroid formation age relative to Eulalia, which has a larger (38%) population of very old 1st-generation bodies that are delivered.

The crater retention ages for Bennu and Ryugu are 10-65 Myr and <30 Myr respectively[8,32,33]. From the suite of model parameters

considered here, the case with highest fraction of ages below 60 Myr and the lowest median age assumed a family SFD with a slope of −3 and a largest fragment size of 0.8 times the largest remnant for the New Polana family. Here the median age was 90 Myr, with 46% and 34% with 60 Myr and 10 Myr, respectively.

Simulated asteroids from the 2nd- or later generations were, by definition, not directly created in the original family breakup and thus have a direct parent smaller than the family parent body. Most $N$th-generation objects (97%) are the largest remnant of a disruption of only a slightly larger body—they are the largest remnants of their direct parent. For the set of nominal parameters, the smallest parent had a diameter of 0.63 km, where the relationship between smallest allowable direct parent depended on the simulation settings (Fig. 6). Increasing the relative size of the largest fragment decreases this fraction as more larger bodies are produced at each disruption and intermediate sizes were subsequently more populated in the distribution. Steeper SFD slopes increase the fraction of objects that are the largest remnants.

## Discussion

The family evolution model presented here, for a wide range of collisional parameters, finds that if both Bennu and Ryugu originate from the New Polana or Eulalia families they are likely 2nd or later generation remnants. Remnants delivered from both families in this size range have median ages since their last disruption that are substantially younger than the family age, where roughly 33% of New Polana members being delivered had their last disruption on timescales similar to Bennu's crater retention age of about 60 Myr, and their direct parents were likely only slightly larger. The very low probabilities for delivered asteroids 0.5 to −1 km to be 1st-generation remnants of families generally fits with the disparity in the collisional lifetimes of these objects (about 200 to −440 Myr) relative to the much older family age (830 to −1400 Myr). This highlights the importance of linking NEAs with families as it provides significant context to their collisional and dynamical histories.

More generally it quantifies the expectation that small asteroids, with sizes similar to those of Ryugu and Bennu, do not live forever[44]. Collisional lifetimes much shorter than Solar System timescales have been estimated from numerous lines of study including hydrodynamic models of catastrophic disruptions[45], models of the collisional evolution of the known orbital and size distribution of asteroids in the Main Belt[7] and from disruption scaling laws built upon measurement of the strengths of boulders found on Bennu's surface[31]. In fact, modeling tools have had notable successes recently in closely matching outcomes of impacts at relevant speeds and energies on actual asteroid surfaces via the Small Carry-on Impactor at Ryugu[46,47] and the Double Asteroid Redirection Test (DART) mission impact on asteroid Dimorphos[48]. Here we found that the dynamical evolution of these small bodies provides another mechanism to shorten their expected lifespan, as they need to avoid both collisional disruption and dynamical escape, finding the median ages of asteroids escaping in this model is shorter than the collisional timescales for km-sized bodies.

The results here have a strong dependence the collisional lifetime for rubble pile asteroids. A simple experiment of lengthening or shortening the lifetime for each asteroid by a factor of 3× resulted in extending and decreasing the crossover time in the fiducial New Polana simulation (1430 Myr and 260 Myr). While this is not a linear dependence, it is correlated and important. Two recent disruption laws built to satisfy different constraints from the Main Belt population estimate collisional lifetimes within a factor of about 2 in the size ranges modeled here[7,31,41]. Thus, within the framework where these works are good estimates for collisional lifetimes, and the results presented here are robust. Meanwhile, while the Yarkovsky drift rates are based on a measured rate for asteroid Bennu, faster or slower orbit drift rates can change the crossover times and the relative balance of 1st, 2nd, and 3rd generation bodies over time, with faster drift rates leading to more 2nd generation fragments and slower drift rates increasing crossover time (Table 2).

## Table 1 | Delivery probabilities

|  | New Polana at 1400 Myr | Eulalia at 830 Myr |
|---|---|---|
| 1st gen | 21% | 38% |
| 2nd gen | 41% | 39% |
| 3rd gen | 22% | 18% |
| 4th gen | 13% | 5% |
| 5th gen | <3% | <1% |

New Polana and Eulalia family probabilities for delivery at 1400 Myr and 830 Myr respectively after family formation with the parameters of the second largest fragment size being 0.4 diameter of the largest remnant, −4.0 cumulative family slope, and a 100 km parent body. Delivery statistics are averaged over the interval 1350–1450 Myr and 780–880 Myr for the two families respectively. The crossover times are at 450 Myr and at 510 Myr respectively for the two families with these parameters.

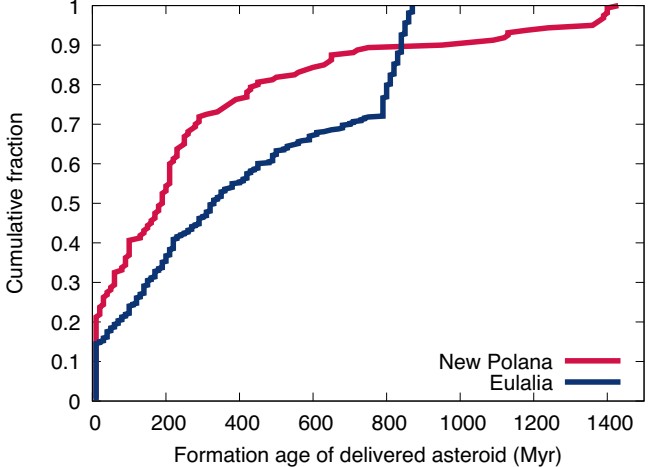

**Fig. 5 | Formation ages of delivered asteroids.** The cumulative fraction of formation ages of the km-size simulated asteroids delivered to the inner solar system from the New Polana and Eulalia families in red and navy respectively, for the nominal set of simulation parameters.

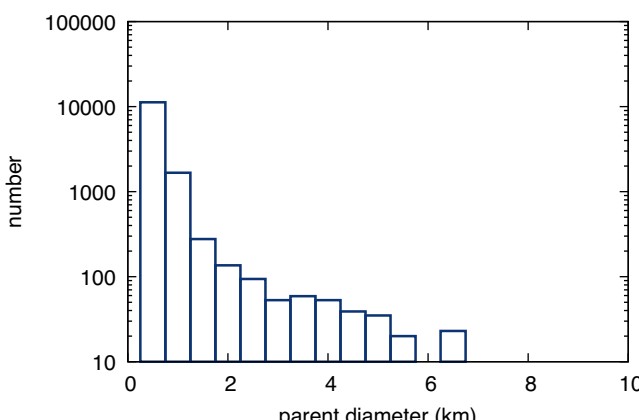

**Fig. 6 | Immediate parent sizes.** The histogram shows the size of the immediate parent for $N$th generation bodies from the New Polana family that reach the $v_6$ or J7:2M5:9 resonances. This simulation used nominal settings simulation with 300,000 initial asteroids.

## Table 2 | Delivery percentages for different drift rates

|          | Fast drift | Nominal | Slow drift |
|----------|-----------|---------|------------|
| 1st gen  | 29%       | 21%     | 22%        |
| 2nd gen  | 49%       | 41%     | 38%        |
| 3rd gen  | 15%       | 22%     | 28%        |
| 4th gen  | 6%        | 13%     | 9%         |
| 5th gen  | 0%        | <3%     | 3%         |

Generation delivery percentages for the Fast, Nominal and Slow drift cases for nominal New Polana family delivery case.

## Table 3 | Delivery fractions for Flora and Baptistina

|          | Flora at 1400 Myr | Baptistina at 160 Myr |
|----------|-------------------|------------------------|
| 1st gen  | <1%               | 77%                    |
| 2nd gen  | 42%               | 21%                    |
| 3rd gen  | 31%               | 2%                     |
| 4th gen  | 20%               | <1%                    |
| 5th gen  | 5%                |                        |

Left: Flora family deliveries for 1400 Myr after family formation with the parameters 0.4 diameter of the largest remnant for the second largest fragment size and −4.0 cumulative family slope for a 200 km parent body. The crossover is at 370 Myr. Right: Baptistina family probabilities for delivery at 160 Myr with the same nominal parameters and a 60 km parent body, where it does not crossover to Nth-generation dominated delivery over the expected age of the family.

The parent bodies of the two asteroid families studied here are expected to be on order about 100 km, and the visited asteroids, Ryugu and Bennu, both represent less than one millionth of the original parent body mass. If they were formed in the original family forming impact (1st-gen) they may sample only a tiny fraction of any diversity of the parent body, particularly any variations with depth. Notably, both Bennu and Ryugu show heterogeneity that has been attributed to their regolith being sourced from materials from distinct zones within the parent body[8,49]. An asteroid that is a 2nd or later generation remnant implies that it experienced further collisional evolution where the impactor for each subsequent breakup event has a chance to further alter original parent body material while potentially polluting the remnants with new and unrelated material. Exogenous material was detected on both asteroids, with V-type material discovered on Bennu[50], and exogenous boulders on Ryugu[51] found to have different space weathering trends that possibly reflect distinct unweathered spectra from presumably different exogenous contributions[52].

A similar mass balance argument may explain the lack of shock features detected among samples from Ryugu so far. The physical properties of Ryugu materials was found to decay the impact-induced shockwave very rapidly due primarily to the very high porosity of the material[53]. This results in minimal volume being affected by the shock of each large impact and a lower probability of any sample carrying the signature of the entire collisional history of the asteroid.

One of the simplifying assumptions of this model is that the only meaningful collisions are precisely at the energy where half of the mass is liberated from the target ($Q_D^*$). Of course, there are more, or less, energetic collisions that will happen, but previous efforts exploring the collisional evolution of the Main Belt size distribution find minimal changes when using more sophisticated accounting of these collisions[34] when compared with more simplistic approaches[7], similar to those adopted in this work. While collisions substantially more energetic than $Q_D^*$ are rare events they could provide pathways to more widespread shock on the target and mixing of material throughout its reaccumulated remnants.

The comparison of crater retention age is done with the formation of a simulated asteroid in a catastrophic collision, but there are likely other ways to reset crater retention ages. The mapping of craters on Ryugu and Bennu[8,32] and the artificial crater made on the surface of Ryugu[46] have found that cratering happens on small rubble piles in a gravity-controlled manner. More simply, craters are larger for a given impactor size than expected from strength-dominated craters, so much so that scaling relationships predict craters larger than the target for impact energies below that expected to disrupt the body. This suggests that there are possibly s sub-catastrophic cratering outcome that are capable of nearly, or entirely, resurfacing (and possibly re-shaping) small rubble piles while not destroying them[54]. The signature of this as a widespread effect would be systematic measured crater retention ages that are less than the formation times for asteroids in this size range.

Similarly, YORP spinup and re-shaping has been attributed as the likely source of the distinct top-shape that both Bennu and Ryugu have and is common among NEAs and especially binary asteroids which account for 15% of the NEA population[55]. YORP spinup is possibly a key or driving element of the spectral changes amongst S-type asteroids where movement of surface material is needed to liberate or expose fresh material that has not been exposed to space weather (see ref. [56]). Therefore, YORP spinup may provide a pathway to re-surface via landslides and re-shaping on timescales shorter than collisional lifetimes[57–61]. The YORP effect may also combine with the effect of tides during close flybys of the terrestrial planets or the effects of seismic shaking due to small impacts, to further contribute to the erasure of some cratering history on small asteroids[62–65].

However, the relatively young retention ages for both asteroids are reasonably matched by the modeled collisional scenarios, particularly for New Polana (21% to −33% chances for a simulated asteroid to have such a young surface). Furthermore, YORP spinup or sub-catastrophic impacts that resurface and reset crater retention ages would then allow for older formation ages of Bennu and Ryugu, closer to the median age of 190 Myr for those delivered from the New Polana family. The probability of either asteroid originating from the Eulalia family is lower with 15 to −19% of delivered bodies having similar formation times and a median age of 330 Myr. Here, a re-surfacing mechanism could provide a path to reconcile the young crater retention ages with expected formation ages of family members.

The shapes of both Bennu and Ryugu (and Didymos, another near-Earth asteroid) are notable for having equatorial ridges and being top-shaped[37,38,66]. One possible origin of these distinct and similar shapes is a result of spinup and re-shaping by the thermal YORP effect[57,58,67,68], which is more effective for near-Earth orbits than for Main Belt asteroids. A recent formation of an equatorial ridge on NEA timescales of a few Myr, is challenged by the notable abundance of large craters in the equatorial regions on Bennu[32,66] suggesting that much of its cratering history post-dates ridge formation[65]. Alternatively, the shapes could have been obtained when the objects reaccumulated, essentially at formation time as an artifact of the reaccumulation process itself[69,70]. A reaccumulation origin would still be compatible with an Nth-generation remnant as each generation relies on a disruption and reaccumulation process, so the re-shaping could occur at the simulated asteroid's formation time in this model. This idea is less compelling if sub-catastrophic disruption is needed to explain a substantially younger asteroid that is linked to the Eulalia family with the longer expected median ages (340 Myr) and their relatively low probability of only 13% to be younger than 10 Myr. Here, resurfacing by sub-catastrophic impacts, may alter or distort the full shape of the asteroid and eliminate the simple and symmetric top-shape[54].

These are not the only rubble piles visited by spacecraft, though others have a less definitive link with specific asteroid families, which is a key simplifying assumption. Both asteroids Itokawa and Didymos have similarly high likelihoods of escaping from the Inner Main Belt by the same two dynamical pathways. The same analysis with identical settings, including the same $Q_D^*$ law despite these being different

taxonomies, was performed assuming that they originate from the Flora and Baptistina families respectively with a larger 200 km parent body for Flora and 60 km for Baptistina[41,63]. Flora has a similar age to New Polana[71] but is much closer to the inner edge of the Main Belt centered at 2.201 au, so for nominal model parameters 99% of km-sized remnants being delivered today are 2nd-generation of later (Table 3). Meanwhile, the Baptistina family is only thought to be approximately 160 Myr and therefore a much greater percentage of deliveries, 77%, were found to be 1st-generation fragments. The potential history of these two asteroids, along with Bennu and Ryugu, show how closely their collisional history is intertwined with their asteroid family of origin and motivates further efforts to link NEAs of interest with their potential ancestors in the main asteroid belt.

## Methods

### Collisional disruption calculations

The code steps forward in 10 Myr time increments. Within each time-step each simulated asteroid is tested, based on its size, to determine if it suffers a disruptive collision. This was calculated with the $Q^*_D$ formulation and coefficients from ref. [31], shown in to be similar to the results in ref. [7] in the size range of interest. Notably ref. [31] disruption law was determined from boulder strengths on Bennu and Ryugu.

Changing the simulation timestep to 5 Myr did not change the crossover point for Eulalia with these settings and New Polana changed the crossover time by <2%.

### Yarkovsky drift rates for simulated asteroids

The semi-major axis drift rates are anchored on the measured drift rate of 1.9e−3 au/Myr measured for Bennu[39,40], and here they are scaled by their semi-major axis, their size ratio and their obliquity (where Bennu's obliquity is nearly 180°).

$$\frac{da}{dt} = \left(\frac{da}{dt}\right)_{bennu} * sqrt\left(\frac{a_{bennu}}{a}\right) * \left(\frac{D_{bennu}}{D}\right) * \cos(obliquity) \quad (1)$$

where $a$ and $D$ are the target bodies semi-major axis and Diameter, $a_{bennu}$ is Bennu's semi-major axis, $D_{bennu}$ is Bennu's diameter and *obliquity* is the target bodies orbital obliquity. The measured rate for Bennu is very accurate owing to the long-duration radio tracking provided by the OSIRIS-REx mission[40], but it is possible that Bennu is not representative of the modeled families or that there is significant variation among members of any given family. Changing the drift rate by a factor of 3 in either direction (faster or slower) does change the outcomes. The faster drifting case had the same crossover time of 470 Myr, but with a strong shift to second generation remnants and a decreased median formation age for delivered objects. Notably there are very few remnants being delivered at 1400 Myr relative to the earlier peak delivery times as the total numbers in the family are heavily depleted due to the fast drift to resonances. The slow drifting case had a moderate shift to 3rd and later generations but a longer crossover time of 720 Myr, after which $N$th-generation fragments dominate.

### Flora and Didymos from Flora and Baptistina families

Itokawa, an LL chondrite[72], is likely to have originated from the Flora family at 2.202au at the innermost edge of the main belt[41], and Didymos is a good match for the Baptistina family in a similar region of the inner Main Belt[63]. Using those families as possible origins, and the same drift rate as Bennu and the rest of the simulation nominal parameters the outcomes strongly support Itokawa as $N$th-generation (95%) and Didymos 1st generation (77%).

### YORP cycles and stochastic YORP

This implementation of the model requires tracking spin rates for each simulated asteroid as change in obliquity is spin-rate dependent. Both

spin rate and obliquity change on timescales as short as ~1 Myr, and thus the simulation timestep was decreased to 0.01 Myr. The functions for obliquity change and spin rate change are those described in refs. [20,26], and collisional reorientation, described in ref. [20] and derived from ref. [73]. These formulations require several additional parameters, briefly described here, but more detailed explanation in ref. [26], that provides numerous fits to the evolution of the Eulalia and New Polana families.

The strength of the YORP effect is controlled by $c_{YORP}$, and was set at 0.6 throughout. $\delta_{YORP}$ controls the bias for spin-up and spin-down following a reorientation, and is set at 0.5. $\tau_{YORP}$ is the timescale for the stochastic YORP process of selecting a new YORP condition for spin rate change but not altering the obliquity. In Bottke et al. 2015 this was done in a size-dependent manner and that is re-created here where for diameter larger than 3 km $\tau_{YORP} = 1$ Myr, for diameters larger than 1.9 km and smaller than 3 km $\tau_{YORP} = 0.5$ Myr and for diameter smaller than 0.9 km $\tau_{YORP} = 0.25$ Myr.

## Data availability

All data were produced with custom computer code with the settings described in the work and can be reproduced with the code from the code repository described in code availability.

## Code availability

The code used for this work is available for download from the following repository[74]. https://github.com/walsh7987/AsteroidFamilyEvolution.

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

## Acknowledgements
This material is based upon work supported by NASA under Award NNH09ZDA007O and Contract NNM10AA11C issued through the New Frontiers Program. We are grateful to the past and present members of the OSIRIS-REx Team who made the return of samples from Bennu possible. This award supported K.J.W., R-L.B., W.F.B., H.C.C., D.N.D., E.R.J., T.M., M.C.N. NASA under Contract 80NSSC22K0045 issued through the Solar System Workings Program supported K.J.W., R-L.B and S.R.S. French space agency CNES supported M.D. and P.M. Support from ANR "ORIGINS" (ANR18-CE31-0014) supported C.A. and M.D. JSPS funding (RECONNECT project 22K21344 & Kakenhi 23H00141) supported S.S and T.M.

## Author contributions
K.J.W. contributed Conceptualization, Funding acquisition, Methodology, Software validation and visualization, Writing – original draft and review and editing. R-L.B., W.F.B. contributed Conceptualization, Methodology, Writing – original draft and review and editing. C.A., H.C.C., M.D., D.N.D., E.R.J., T.M., P.M., T.M., M.C.N., S.R.S., S.S., D.S.L., contributed Conceptualization and Writing - review and editing.

## Competing interests
The authors declare no competing interests.
