## [Peer Review File · Nature Communications]

REVIEWER COMMENTS

Reviewer #1 (Remarks to the Author):

Authors: Walsh et al.,

Title: Bennu and Ryugu likely 2nd or later generation rubble piles

Rubble pile asteroids are a subject of interest. They represent an aggregation of debris formed in catastrophic disruptions between "parent" asteroids. Each collision of this kind has the potential to give rise to an asteroid family. This study employs models of collisional and dynamical evolution of at least two distinct asteroid families (New Polana and Eulalia). The aim of the study is to determine the probability of asteroids Ryugu & Bennu being direct remnants ("1st generation") bodies, from the initial disruption event that created their respective families. Alternatively, it seeks to ascertain if these asteroids have undergone extensive evolution and experienced multiple disruptive events over their lifetime.

The Near-Earth asteroids Ryugu and Bennu hold particular interests to the wider audience thanks to the samples retrieved from their surfaces by JAXA's Hayabusa2 (from Ryugu) and NASA's OSIRIS-REx (from Bennu). This research is timely as the sample collected from Bennu is scheduled to reach Earth on September 24th.

The manuscript is well written and layed out in a logical manner. However, I do raise some concerns regarding the assumptions made for the modeling work and the implications of the presented work. Below is a list of "issues" requiring substantial attention from the authors.

In the calculations for collisional disruption, the authors have relied on the Q^*D laws as derived by Ballouz et al. (2020). It's worth noting that these scaling relationships have been derived from fitting to measurements of cratered and cracked boulders on Bennu's surface. While this approach may be suitable for the initial parent body, which can be treated as a monolith, it may not hold true for subsequent impacts on modified asteroids that may themselves be rubble piles. This raises the question of how the results would be affected if one were to assume a different Q^*D curve for the subsequent collisions. Additionally, I'm curious about the robustness of the presented results in light of the choice in Q^*D curves.

The implications of this study remain somewhat ambiguous, lacking a definitive argument for its novelty. For example, on line #110, the authors suggest implications for interpreting the returned sample; however, this point is not elaborated beyond the introduction. Given that samples from Ryugu have already undergone analysis and resulted in numerous published papers, it begs the question of whether any parallels can be drawn.

At line #485, the authors have extended the same analysis used for C-type asteroids Ryugu and Bennu to S-type asteroids Itokawa and Didymos. Nevertheless, there is presently no compelling evidence to suggest that the Q*D equations would be applicable to the less porous S-type asteroids. Therefore, additional explanation is needed or the paragraph removed altogether.

#102: Some references are needed

#186: There should be a comma after (Bottke et al. 2022), not full stop.

There are a couple of repetitions throughout the manuscript. See Lines #111 & #351

Reviewer #2 (Remarks to the Author):

The manuscript "Bennu and Ryugu likely 2nd or later generation rubble piles" investigates a scenario in which both asteroids are 2nd or later generation fragments from an inner asteroid belt body. The study is motivated by apparently very different ages of asteroid families where the asteroids are supposed to originate from and the crater retention ages. To this purpose, the authors employed a novel, Monte-Carlo-based approach that simultaneously treats the collisional and dynamical evolution. This is one of the first works in this respect and is appropriate for the problem studied.

Indeed, the model is based on different assumptions and approximations, but the authors also investigated how sensitive the results are concerning some of the input parameters. For instance, the authors show that the results do not change significantly for different slopes of the size-frequency distribution nor when varying the sizes of the parent bodies.

The manuscript is well-written, and the figures and tables are appropriate and informative.

Therefore, it fully deserves publication.

There are only minor issues to be addressed:

Generally, if I'm not missing something, it is not described when exactly a test particle is assumed to be entering the NEAs region, i.e., when it is considered to be 'delivered'.

The ν_6 and $3/1$ resonances have been supposed to remove 100% of objects, right? But what happens to them? I assume that a fraction from each resonance is considered to be delivered into the NEA region, but I cannot find precise information. This aspect of the model should be better described, and missing details should be provided.

line 41: "All but the largest NEAs are expected to be rubble piles"

This sounds like a too bold statement. Are you claiming that also all objects below ~ 100 m are rubble piles? Even super-fast rotators? Please rephrase and/or be more specific.

line 54: "Models of NEA delivery find that the probability of a body on Bennu's current orbit has a 82% and a 18% probability to originate from the ν_6 or IMC, respectively"

It would be helpful to provide here also the probabilities and routes from the model by Granvink et al. (Icarus, 2018)

line 61: Nesvorny et al. 2015 is definitely an appropriate reference for asteroid families.

Yet, it might be good to cite along a more recent review by Novakovic et al. CeMDA, 2022, which is some aspect complementary.

line 157, Figure 1: The reference for the family ages should be added, Bottke et al. (Icarus, 2015)

line190: "There is some initial dispersal of orbital semi-major axis that scales with the escape velocity of the target bodies (see Vokrouhlicky et al. 2006), but here the very long timescale of their evolution overwhelms the initial scatter of initial orbital semi-major axis, so each family member starts with identical semi-major axes."

I agree that the initial dispersions should play a minor role compared to post-impact evolution. It seems,

however, that it would be simple to implement this aspect into the model. At least for larger parents, it might be relevant because the expected ejection velocities are higher. Also, in the case of Eulalia, which is close to the 3/1 resonance, adding the initial dispersion might place some bodies on the other side of the resonance.

line 244: "Note that the removal efficiency is drift rate-dependent, which itself is size dependent (Bottke et al. 2015) and these removal calculations are valid only for the km-scale objects."

Milic Zitnik & Novakovic (ApJ, 2016) studied the effect of some MMRs on the semi-major axis drift due

to the Yarkovsky effect. Though it may not be directly relevant to this work, it could be worth

mentioning it here as well, as, in general, this analysis could be very useful in the model like the one presented in the manuscript.

line 379, Figure 6: This figure's caption says there were 300,000 initial asteroids. Was that the number of test particles in all simulations? Please specify. Also, this number should be stated earlier when the simulation set-up is described.

line 485: "Flora has a similar age to New Polana..."

Please add a reference for the age of the Flora family, Vokrouhlicky et al. (AJ, 2017)

line 513: In the Yarkovsky scaling relation, it seems that the scaling with the obliquity of Bennu is missing.

Or is it omitted because it is close to 180 deg, implying that cosine is close to -1, and the sign is neglected?

Please check and fix or add missing details.

Reviewer #3 (Remarks to the Author):

Main belt asteroids can be disrupted by catastrophic collisions, leading to the formation of fragments. These fragments are referred to as a "family" of the original asteroid. The orbits of these fragments, now smaller asteroids, can be altered by the Yarkovsky effect. Additionally, these fragments can be further disrupted by subsequent collisions with other small bodies. When a main belt asteroid fragment reaches a certain radial location, known as an orbital resonance, it can be dynamically transferred to the near-Earth region.

This study utilized numerical simulations to examine the orbit evolution of small bodies driven by the Yarkovsky effect. It combined this with the catastrophic collisional evolution using a Monte-Carlo approach. The authors aimed to investigate specific asteroid families, such as New Polana and Eulalia, and to discuss their collisional and orbital evolution. Recently, spacecrafts have visited near-Earth asteroids Ryugu and Bennu. These asteroids are thought to be members of the collisional families of New Polana or Eulalia. However, the surface ages of these asteroids, as reported, are significantly younger than the estimated age of their parent family's formation. Through numerical simulations, the authors determined that Ryugu and Bennu are not fragments from the initial family-forming collision. Instead, they are products of subsequent, additional collisions.

In my assessment, the findings presented are both intriguing and relevant to the current scientific context. The manuscript is largely well-written and the content is effectively conveyed. As such, I believe it meets the standards for publication in Nature Communications. However, before I can recommend its acceptance, I would like the authors to provide further clarity on their numerical models. Additionally, I urge them to discuss the consistency of their findings with the returned samples from Ryugu — and from Bennu, if available. Addressing these points is crucial for validating and bolstering the credibility of their numerical results.

Major comments.

(1) In lines 125-130: I would appreciate it if the authors could first verify whether their new approach — encompassing both orbital and collisional evolution — are able to accurately reproduce the Size-Frequency Distribution (SFD) of asteroids within the main asteroid belt. Additionally, could the authors discuss any updates or differences in their findings, especially in comparison to the results presented in Bottke et al. 2005 and Morbidelli et al. 2009?

(2) In line 415, the authors note that collisional evolution might significantly modify the parent body materials, leading to a mixture of materials from both the impactor and the target. However, recent literature, specifically Nakamura T. et al. 2022 in Science, has indicated that sample material from Ryugu doesn't show evidence of impact shock. Furthermore, Tatsumi et al. 2021 reported that the fraction of exogenic materials in the Ryugu sample is comparatively small. One would anticipate that multiple catastrophic impacts would both shock the materials and promote mixing between the impactor and target. This presumption seems to contradict the sample analysis results from Ryugu, as mentioned earlier. Could the authors provide an explanation for this apparent discrepancy?

In line with the aforementioned concerns, I request the authors to provide specific details regarding the impact conditions encountered during the collisional evolutions. Can the authors elucidate on the impact velocity and the impactor-to-target size ratio as derived from their simulations? Following

this, I would appreciate it if the authors could estimate the consequent increase in pressure and temperature resulting from the specified impact conditions. I hypothesize that if the impactor-to-target size ratio is close to unity, it would likely lead to enhanced material mixing. Could the authors comment on this?

(3) In lines 185-190: The authors have posited that collisions with background asteroids primarily influence family members. However, immediately following the initial catastrophic collision that results in the formation of an asteroid family, the fragments would likely have orbits in close proximity to one another, given that they would share an almost identical point of impact. Consequently, right after the impact, there's a plausible scenario where these fragments (or family members) might collide with each other frequently, leading to further fragmentation. Could the authors provide a discussion or estimation on the collision timescale under such conditions?

Minor comments:

Hirabayashi et al. 2020; Hyodo & Sugiura 2022 also discuss the shape change and resurfacing of small asteroids.

To whom it may concern,

Below we endeavor to answer all of the comments of the two reviewers and describe or list the changes that we made to the manuscript. We believe that we adequately answer all of the concerns and that the main findings of this work are unchanged. You will find the Reviewers comments in *Italics* and our responses in normal font.

Reviewer 1:

Rubble pile asteroids are a subject of interest. They represent an aggregation of debris formed in catastrophic disruptions between "parent" asteroids. Each collision of this kind has the potential to give rise to an asteroid family. This study employs models of collisional and dynamical evolution of at least two distinct asteroid families (New Polana and Eulalia). The aim of the study is to determine the probability of asteroids Ryugu & Bennu being direct remnants ("1st generation") bodies, from the initial disruption event that created their respective families. Alternatively, it seeks to ascertain if these asteroids have undergone extensive evolution and experienced multiple disruptive events over their lifetime.

The Near-Earth asteroids Ryugu and Bennu hold particular interests to the wider audience thanks to the samples retrieved from their surfaces by JAXA's Hayabusa2 (from Ryugu) and NASA's OSIRIS-REx (from Bennu). This research is timely as the sample collected from Bennu is scheduled to reach Earth on September 24th.

The manuscript is well written and layed out in a logical manner. However, I do raise some concerns regarding the assumptions made for the modeling work and the implications of the presented work. Below is a list of "issues" requiring substantial attention from the authors.

*In the calculations for collisional disruption, the authors have relied on the Q^*D laws as derived by Ballouz et al. (2020). It's worth noting that these scaling relationships have been derived from fitting to measurements of cratered and cracked boulders on Bennu's surface. While this approach may be suitable for the initial parent body, which can be treated as a monolith, it may not hold true for subsequent impacts on modified asteroids that may themselves be rubble piles. This raises the question of how the results would be affected if one were to assume a different Q^*D curve for the subsequent collisions. Additionally, I'm curious about the robustness of the presented results in light of the choice in Q^*D curves.*

We thank the reviewer for these very good comments. We address them in turn.

First, the dependence of our results on Q^*_D curves is a very interesting question. We note that the Ballouz et al. (2020) Q^*_D scaling relationship transitions from strength- to

gravity-dominated at an asteroid diameter of 160 m (Otohime Saxum's diameter). The gravity-dominated part of the Ballouz et al. (2020) is derived from observations of craters on large asteroids and scaling constants derived from the hydrocode disruption studies of Leinhardt & Stewart (2012). Leinhardt & Stewart (2012) compiled data from catastrophic-disruption simulations from various sources and found that for (i) a variety of target compositions, (ii) five orders of magnitude in size ($\sim 0.4\text{--}4e5$ km) and (iii) nine orders of magnitude in impact energy, the Q^*_D relationship could be described by a narrow range in scaling constants. Therefore, the Q^*_D disruption relationship would be applicable to gravity-dominated rubble piles regardless of their composition. Note that Q^*_D relationships describe a *specific* impact energy. As it is detailed later in our response to reviewers (page 5 - paragraph starting with "Indeed, Leinhardt & Stewart (2012) compiled data...") it is understood that C- and S-type might have very similar Q^*_D relationships in the gravity regime (rubble piles), so a C-type would indeed require less energy to catastrophically disrupt than a S-type of the same size, due to purported differences in their bulk densities.

The work presented here really only deals with collisions in the size range of 0.5km-~10km, where the vast majority of the collisions are happening at the small end. In this range the expected collisional lifetime between Bottke et al. (2005) and Ballouz et al. (2020) differs by no more than a factor of $\sim 2x$ - with Bottke's yielding a shorter lifetime. Also note that the recent scaling law from Bottke et al. (2020) is even closer to Ballouz et al. (2020). We did a series of test runs where we simply use a 3x larger or smaller lifetime than the nominal Ballouz2020 case. The results are mostly intuitive - they definitely change the timescales for crossover times - and we put a short description of this in the Discussion section. Here, though we summarize - the simple test we did was to re-run with the Q^* law as formulated and simply use a multiplicative factor of 3x to lengthen or shorten the collisional lifetime from the calculations. For the fiducial Polana case this lengthens the crossover time to 1430Myr or shortens it to 260Myr. The nominal value was 450Myr, so this finds that the trend is clear, but not simple and linear. The 3x lifetime case this increase in lifetime means that 1km asteroids can survive 1.2Gyr and 0.5km can survive for 600Myr.

We addressed this directly in the Discussion with a new paragraph:

"The results here have a strong dependence on the collisional lifetime for rubble pile asteroids. A simple experiment of lengthening or shortening the lifetime for each asteroid by a factor of 3x resulted in extending and decreasing the crossover time in the fiducial New Polana simulation (1430Myr and 260Myr). While this is not a linear dependence, it is correlated and important. Two recent disruption laws built to satisfy different constraints from the Main Belt population estimate collisional lifetimes within a factor of ~ 2 in the size ranges modeled here

(Bottke et al. 2005, Ballouz et al. 2020). Thus, within the framework where these works are good estimates for collisional lifetimes, the results presented here are robust. If new disruption laws emerge from future works these results very roughly scale as do the collisional lifetimes.”

Ref:

Leinhardt, Z. M. & Stewart, S. T. Collisions between gravity-dominated bodies. I. Outcome regimes and scaling laws. *Astrophys. J.* 745, 79 (2012).

The implications of this study remain somewhat ambiguous, lacking a definitive argument for its novelty. For example, on line #110, the authors suggest implications for interpreting the returned sample; however, this point is not elaborated beyond the introduction. Given that samples from Ryugu have already undergone analysis and resulted in numerous published papers, it begs the question of whether any parallels can be drawn.

Fair point - we should have included more direct links to the published results from Ryugu. In fact, its really not as simple as asserted in the initial manuscript and so it has been valuable to add more.

Fundamentally the lack of shock features in Ryugu samples is very interesting. It raises a general question on how Ryugu materials have avoided receiving substantial shock metamorphism during its collisional evolution from a 100-km-sized parent body to the current size. However, we don't think this limits the number of large-scale impacts their various parent bodies have witnessed, which is the main focus of this study. If a meteorite or sample is shocked, it means the parent was shocked. However, the lack of shock on a single meteorite or sample does not necessarily mean that there was no shock and no impacts on the parent. It just means that this meteorite or sample wasn't affected by it due to lack of proximity or other effects. The samples are just not perfect tracers of collisional history. We have tried to clarify this in the text, and we include substantially more text below for the reviewers.

So, there are a couple of factors that could account for the lack of shock features in Ryugu's samples. First is that the mass fraction of impact-heating in the parent body is estimated to be very small. Nakamura et al. (2022) conducted systematic impact calculations using physical properties of Ryugu materials measured using the actual returned samples and showed that the impact-induced shockwave decays so quickly that it should not leave strong shock features in most of the target volume. Fig. S47 caption in that paper's Supplementary Materials indicates that the heated region up to >90 °C is limited within the distance of ~2 impactor diameters from the impact point.

This volume is ~3% of parent body. If we take 250 °C, which is the lowest stage of thermal metamorphism for carbonaceous chondrites (Nakamura et al. 2005) as a threshold, mass fraction predicted from Fig. S47 (Supplementary Materials) becomes even smaller (i.e., <1%). Such rapid shockwave decay and small shock-heated volume is caused by the high porosity nature of Ryugu materials. If Ryugu materials are from 99% or 97% of unshocked volume as T. Nakamura et al. (2022) suggests, then we will might expect to find little shocked materials in the samples.

Second, highly shocked materials generally might be expected to receive higher ejection velocities. This would help shocked materials escape from the impact site. This would further reduce the mass fraction of shocked materials in re-accumulated fragments loosely bound with their weak self-gravity.

Furthermore, regarding the small mass fraction of impactor materials on Ryugu, some of this team is writing a paper on the effect of oblique impacts (Sugiura et al. in prep). Preliminary results indicate that impacts with 45-deg or higher measured from the normal would result in impact mass mixing ratio as low as Ryugu ($\sim 10^{-6}$) Sugita et al. (2023) (LPSC abstract). Thus, small apparent mass mixing ratio observed by Sugimoto et al. (2021a) is not a problem.

All that being said, we added a bit of extra content at the first mention around line 110, and then a short paragraph in the Discussion to expand on these topics - particularly the mass balance and the samples potential low likelihood of carrying shock signatures.

“A similar mass balance argument may explain the lack of shock features detected among samples from Ryugu so far. Nakamura et al. 2023 found that the physical properties of Ryugu materials decays the impact-induced shockwave very rapidly due primarily to the very high porosity of the material. This results in minimal volume being affected by the shock of each large impact and a lower probability of any sample carrying the signature of the entire collisional history of the asteroid.”

*At line #485, the authors have extended the same analysis used for C-type asteroids Ryugu and Bennu to S-type asteroids Itokawa and Didymos. Nevertheless, there is presently no compelling evidence to suggest that the Q*D equations would be applicable to the less porous S-type asteroids. Therefore, additional explanation is needed or the paragraph removed altogether.*

This is an interesting point, and we understand the concern. Unfortunately, to our knowledge, there also is no definitive data yet available that shows that S-type asteroids have a different Q^*_D than C-types for asteroids the sizes of Bennu/Ryugu. The limited information we have instead suggests they may be similar.

For example, from the numerical hydrocode side of things, the closest work on this might be the Jutzi et al. (2019) paper introducing and describing the P-alpha porosity model used for ~100km scale impact modeling, where they state this in their abstract: “Overall, the porous targets have a significantly higher impact strength (Q^*_D) than the rubble-pile parent bodies investigated previously (Benavidez et al., 2012) and show a behavior more similar to non-porous monolithic targets (Durda et al., 2007).” But this work is specifically describing findings of pore-crushing effects on monolithic targets, whereas with rubble piles much of the impact energy might be expended in simply gravitational dispersion of the target, which would seemingly level the playing field between different material properties.

Indeed, Leinhardt & Stewart (2012) compiled data from catastrophic-disruption simulations in the gravity regime from various literature sources and found that for (i) a variety of target compositions, (ii) five orders of magnitude in size (~0.4–4e5 km) and (iii) nine orders of magnitude in impact energy, the Q^*_D relationship could be described by a narrow range in scaling constants. Therefore, the Q^*_D disruption relationship would be applicable to gravity-dominated rubble piles regardless of their composition. Note that Q^*_D relationships describe a specific impact energy, so a C-type would require less energy to catastrophically disrupt than a S-type of the same size, due to purported differences in their bulk densities. On the other hand, C-types have a larger porosity than S-types, and simulations by Martin Jutzi and others suggest this could make them more difficult to disrupt. The two effects combined seem to compensate for one another, at least in hydrocode impact simulations, and this suggests that S- and C-types may have similar Q^*_D functions.

A second example would be to consider cratering among main belt asteroids, which is related to Q^*_D . Ideally, the production of a big enough crater on a world should disrupt the target, so cratering and disruptions have to be self consistent with each other. Bottke et al. (2020) modeled the collisional evolution of the main belt size frequency distribution (SFD) using a 1-D collisional evolution code and derived Q^*_D function that could match constraints. Given that ~ 80% of the main belt is C-complex, it makes sense that their Q^*_D would be most suitable to those bodies. What they found is that their derived SFD did a good job reproducing the observed crater SFDs on all $D > 10$ km asteroids, including those which were S-types (e.g., Eros, Gaspra, Ida), C-types (e.g., Ceres, Mathilde), and other types (e.g., M-type Lutetia, V-type Vesta). So, if big Q^*_D differences existed between different asteroid types, one would probably expect to

see a modeling failure when attempting to match crater SFDs on different kinds of bodies.

The bottom line is that we believe there is no simple “correct” answer in the literature to address this adequately, so instead we made an effort to more strongly caveat our findings on this point. Part of the additional caveats and language added is the clarification that essentially all of the targets in this model are rubble piles as opposed to monoliths.

We have also added an entirely new paragraph to the Discussion to answer a different reviewer’s comment, explaining that the results here roughly scale with collisional lifetime - so a reader now has this additional context regarding lifetimes and Q^*_D formulations.

Ref:

Jutzi, M., et al. (2019) Icarus 317, 215.

Bottke, W. F. and 14 colleagues 2020. The Astronomical Journal 160.

#102: Some references are needed

Added

#186: There should be a comma after (Bottke et al. 2022), not full stop.

Fixed

There are a couple of repetitions throughout the manuscript. See Lines #111 & #351

Yes, this was a serious repetition that has been corrected. We cut this sentence on its second appearance:

”This is much younger than the age of the two potential source families and younger than the collisional lifetime for asteroids their size (~200 Myr and ~440 Myr respectively). “

Reviewer #2 (Remarks to the Author):

The manuscript "Bennu and Ryugu likely 2nd or later generation rubble piles" investigates a scenario in which both asteroids are 2nd or later generation fragments

from an inner asteroid belt body. The study is motivated by apparently very different ages of asteroid families where the asteroids are supposed to originate from and the crater retention ages.

To this purpose, the authors employed a novel, Monte-Carlo-based approach that simultaneously treats the collisional and dynamical evolution. This is one of the first works in this respect and is appropriate for the problem studied.

Indeed, the model is based on different assumptions and approximations, but the authors also investigated how sensitive the results are concerning some of the input parameters. For instance, the authors show that the results do not change significantly for different slopes of the size-frequency distribution nor when varying the sizes of the parent bodies.

The manuscript is well-written, and the figures and tables are appropriate and informative.

Therefore, it fully deserves publication.

There are only minor issues to be addressed:

Generally, if I'm not missing something, it is not described when exactly a test particle is assumed to be entering the NEAs region, i.e., when it is considered to be 'delivered'. The nu6 and 3/1 resonances have been supposed to remove 100% of objects, right? But what happens to them? I assume that a fraction from each resonance is considered to be delivered into the NEA region, but I cannot find precise information. This aspect of the model should be better described, and missing details should be provided.

Thanks for this - yes, it should have been more clear in the manuscript. Fundamentally, the setup of the model allows an indifference to the absolute or relative delivery rate from the respective resonances. The works that link the two asteroids to these two families (Eulalia and Polana; e.g. Bottke2015) already account for that level of detail and the correct dynamical mixture of the near-Earth population. Here, we simply get to use that knowledge of the dynamical pathway (nu6) and determine that if an asteroid comes from the nu6, and its parent was Polana...then *this* is its likely collisional history. We don't need to compare it to all of the other NEAs from all of the dynamical pathways because we *know* where it came from already. Here we are just reporting on the likely collisional history of simulated objects that we model to follow the same dynamical path. We did not establish this clearly in the original manuscript, but have made the following edits - a whole new paragraph on these important details:

“Furthermore, while the nu6 and 3:1 are effectively 100% efficient at removing asteroids from the asteroid belt, neither are as efficient at placing them onto near-Earth orbits, as some are ejected from the Solar System or achieve some other fate. Here, since previous modeling has found Bennu and Ryugu’s likely dynamical pathway to their current orbits, there is no need to blend the relative probability of NEO delivery between these two removal sources. Instead, we are exploring the collisional history of objects that follow this same dynamical path. Since this effort relies on an Monte Carlo approach the total number of initial simulated asteroids in a family can be as large as computationally feasible as long as it adheres to the family size distribution. The numbers and sizes of fragments made in each subsequent collision and disruption must follow the same rules and make proportionally similar numbers, while still adhering to the family size distribution. Thus, nominal simulations below begin with 300,000 simulated asteroids, although the number grows throughout each simulation depending on the number and nature of the collisions in each.”

line 41: "All but the largest NEAs are expected to be rubble piles"

This sounds like a too bold statement. Are you claiming that also all objects below ~100 m are rubble piles? Even super-fast rotators? Please rephrase and/or be more specific.

Good point - we are definitely NOT speaking about the D<200m objects, so we will revise.

It now reads: **“NEAs larger than a few hundred meters and smaller than 5-10km** are expected to be rubble piles, which are the reaccumulated debris from the collisional fragmentation of larger asteroids “

line 54: "Models of NEA delivery find that the probability of a body on Bennu's current orbit has a 82% and a 18% probability to originate from the v6 or IMC, respectively"

It would be helpful to provide here also the probabilities and routes from the model by Granvik et al. (Icarus, 2018)

Here is what we find on the Lowell website that hosts the data from the Granvik works (Moskovitz et al. 2022 - now referenced in the paper):

Bennu: 86% nu6 and 3% from Hungaria

Ryugu: 88% and 5%

Didymos: 82% and 7%

Itokawa: 88% and 4%

And here is the text that was added:

“Updated NEO models, using a larger number of dynamica sources find similarly high probabilities of delivery from v_6 resonance where Bennu is 86% likely and Ryugu 88% (Granvik et al. 2018; Moskovitz et al. 2018). “

line 61: Nesvorny et al. 2015 is definitely an appropriate reference for asteroid families. Yet, it might be good to cite along a more recent review by Novakovic et al. CeMDA, 2022, which is some aspect complementary.

Thanks for this. We had not seen the new review - and its good to see because a fair bit has happened since 2015 that is now included.

line 157, Figure 1: The reference for the family ages should be added, Bottke et al. (Icarus, 2015)

Added.

line190: "There is some initial dispersal of orbital semi-major axis that scales with the escape velocity of the target bodies (see Vokrouhlicky et al. 2006), but here the very long timescale of their evolution overwhelms the initial scatter of initial orbital semi-major axis, so each family member starts with identical semi-major axes."

I agree that the initial dispersions should play a minor role compared to post-impact evolution. It seems, however, that it would be simple to implement this aspect into the model. At least for larger parents, it might be relevant because the expected ejection velocities are higher. Also, in the case of Eulalia, which is close to the 3/1 resonance, adding the initial dispersion might place some bodies on the other side of the resonance.

The reviewer was totally right here, the initial velocity dispersion from the initial family breakup makes a difference. We implemented this and the re-run for the nominal Polana case with initial velocity dispersion gets a crossover of 450Myr, instead of 470Myr that was reported for not having velocity dispersion. That is ... pretty small but the Eulalia numbers drops a bit more down to 510Myr, and that is significant at the ~10% level. We re-ran everything and updated all the plots and numbers throughout the text. The big picture takeaway is not changed, but some key numbers definitely changed enough to make this worthwhile.

We added a bit to the text:

“There is some initial **size-dependent** dispersal of orbital semi-major axis that scales with the escape velocity of the target bodies **and this is included for each initial family and subsequent sub-family following the formulations of Vokrouhlicky et al. (2006).** “

And later to clarify that each of the sub-families, uses the same formulations:

“Similarly, no bodies smaller than 0.5 km are added. Newly formed bodies are started near the location of their recently disrupted “parent”, **whose size sets the scaling for the the size-dependent spread in the semi-major axis relative to the parent’s** “

line 244: "Note that the removal efficiency is drift rate-dependent, which itself is size dependent (Bottke et al. 2015) and these removal calculations are valid only for the km-scale objects."

Milic Zitnik & Novakovic (ApJ, 2016) studied the effect of some MMRs on the semi-major axis drift due to the Yarkovsky effect. Though it may not be directly relevant to this work, it could be worth mentioning it here as well, as, in general, this analysis could be very useful in the model like the one presented in the manuscript.

Thanks for this - reference added.

line 379, Figure 6: This figure's caption says there were 300,000 initial asteroids. Was that the number of test particles in all simulations? Please specify. Also, this number should be stated earlier when the simulation set-up is described.

Good point, we should have been more detailed in the code setup description. We have added this in the main text in an extra paragraph that clears up a handful of missing details about the modeling, including this.

“Furthermore, while the nu6 and 3:1 are effectively 100% efficient at removing asteroids from the asteroid belt, neither are as efficient at placing them onto near-Earth orbits, as some are ejected from the Solar System or achieve some other fate. Here, since previous modeling has found Bennu and Ryugu’s likely dynamical pathway to their current orbits, there is no need to blend the relative probability of NEO delivery between these two removal sources. Instead, we are exploring the collisional history of objects that follow this same dynamical path. Since this effort relies on an Monte Carlo approach the total number of initial simulated asteroids in a family can be as large as computationally feasible as long as it adheres to the family size distribution. The numbers and sizes of

fragments made in each subsequent collision and disruption must follow the same rules and make proportionally similar numbers, while still adhering to the family size distribution. Thus, nominal simulations below begin with 300,000 simulated asteroids, although the number grows throughout each simulation depending on the number and nature of the collisions in each “

line 485: "Flora has a similar age to New Polana..."

Please add a reference for the age of the Flora family, Vokrouhlicky et al. (AJ, 2017)

Added.

line 513: In the Yarkovsky scaling relation, it seems that the scaling with the obliquity of Bennu is missing. Or is it omitted because it is close to 180 deg, implying that cosine is close to -1, and the sign is neglected? Please check and fix or add missing details.

Its the latter - Bennu is very close to 180deg, so its built-in to the formulations for the reasons you say. We added a short phrase telling the reader this - since its important!

Reviewer #3 (Remarks to the Author):

Main belt asteroids can be disrupted by catastrophic collisions, leading to the formation of fragments. These fragments are referred to as a "family" of the original asteroid. The orbits of these fragments, now smaller asteroids, can be altered by the Yarkovsky effect. Additionally, these fragments can be further disrupted by subsequent collisions with other small bodies. When a main belt asteroid fragment reaches a certain radial location, known as an orbital resonance, it can be dynamically transferred to the near-Earth region.

This study utilized numerical simulations to examine the orbit evolution of small bodies driven by the Yarkovsky effect. It combined this with the catastrophic collisional evolution using a Monte-Carlo approach. The authors aimed to investigate specific asteroid families, such as New Polana and Eulalia, and to discuss their collisional and orbital evolution. Recently, spacecrafts have visited near-Earth asteroids Ryugu and Bennu. These asteroids are thought to be members of the collisional families of New Polana or Eulalia. However, the surface ages of these asteroids, as reported, are significantly younger than the estimated age of their parent family's formation. Through numerical simulations, the authors determined that Ryugu and Bennu are not fragments

from the initial family-forming collision. Instead, they are products of subsequent, additional collisions.

In my assessment, the findings presented are both intriguing and relevant to the current scientific context. The manuscript is largely well-written and the content is effectively conveyed. As such, I believe it meets the standards for publication in Nature Communications. However, before I can recommend its acceptance, I would like the authors to provide further clarity on their numerical models. Additionally, I urge them to discuss the consistency of their findings with the returned samples from Ryugu — and from Bennu, if available. Addressing these points is crucial for validating and bolstering the credibility of their numerical results.

Major comments.

(1) In lines 125-130: I would appreciate it if the authors could first verify whether their new approach — encompassing both orbital and collisional evolution — are able to accurately reproduce the Size-Frequency Distribution (SFD) of asteroids within the main asteroid belt. Additionally, could the authors discuss any updates or differences in their findings, especially in comparison to the results presented in Bottke et al. 2005 and Morbidelli et al. 2009?

We understand the question/desire here, but find this to be well modeled in previous works like Bottke et al. 2005 and Morbidelli et al. 2008 that consider the collisional evolution of the *entire* asteroid belt over very long timescales. Such models necessarily consider *all* of the collisions for *all* of the asteroid belt that will globally average out to match the global asteroid belt SFD. Here, we are considering specific, singular, families that represent a very small fraction of the asteroid belt. This is actually why we explored such a wide range of family SFD - to bracket a huge range of possibilities and show that the selection of SFD is largely unimportant to the specific outcome here. So, we will add more text explaining why we didn't try to take on the entire asteroid belt all at once, but provide here for the reviewer a more verbose explanation.

The entire belt is a mixture of numerous families of different size and age (see Broz et al. 2013, or Nesvorný et al. 2015). The entire catalog of families is uncertain, especially at older ages, and the sizes and ages of the clearly identified families are also both uncertain. The combined asteroid belt SFD is a combination of all of these families of differing size and age and all background asteroids that may not have any history with any family. The impact simulations of Durda 2004/2007 and Michel 2001/2003 find that an initial family SFD varies with the different possible impact properties (impact energy and angle etc.), and collisional modeling of the belt over time (Bottke 2005, see also Vokroulický et al. 2006 for an amazingly detailed work) find that a families' SFD will

change over time due to the collisional evolution imparted by its interaction with the neighboring asteroid belt. So, collision modeling tells us that it's hard or impossible to *a priori* know a family's initial SFD and we also know their SFD will change over time. We also don't know precisely when any family formed, or precisely how large it was when it did form (this is why we explored such a wide range in possible parent body sizes). So, trying to model the model belt over its whole history is a big task with too many uncertain parameters. This is exactly why we did this family-by-family with such wide, enveloping, ranges of parameters - to be sure that we bracketed the range of possible outcomes.

The point is well-taken though - we need to be more explicit about the motivation for the approach that we took. We have added text to be more clear and to more clearly differentiate this approach from the Bottke/Morbidelli SFD modeling approaches. "Other models have developed sophisticated routines to capture the effects of collisional evolution, and its large-scale effects on the size frequency distribution (SFD) of the asteroids in the main asteroid belt (Bottke et al. 2005, Morbidelli et al. 2009). **These large models cover such large timescales and large swaths of the Main Belt that they don't take into account individual asteroid families and don't incorporate the ability to track individual asteroids and their changing orbits.**"

Refs:

Broz M., Morbidelli, A., Bottke, W.~F., Rozehnal, J., Vokrouhlicky, D., Nesvorny, D. 2013. *Astronomy and Astrophysics* 551.

Durda, D.~D. and 6 colleagues 2007. *Icarus* 186, 498–516.

Durda, D.~D. and 6 colleagues 2004. *Icarus* 170, 243–257.

Michel, P., Benz, W., Tanga, P., Richardson, D.~C. 2001. *Science* 294, 1696–1700.

Michel, P., Benz, W., Richardson, D.~C. 2003. *Nature* 421, 608–611.

Vokrouhlicky D., Broz, M., Bottke, W.~F., Nesvorny, D., Morbidelli, A. 2006. *Icarus* 182, 118–142.

(2) In line 415, the authors note that collisional evolution might significantly modify the parent body materials, leading to a mixture of materials from both the impactor and the target. However, recent literature, specifically Nakamura T. et al. 2022 in Science, has indicated that sample material from Ryugu doesn't show evidence of impact shock. Furthermore, Tatsumi et al. 2021 reported that the fraction of exogenic materials in the Ryugu sample is comparatively small. One would anticipate that multiple catastrophic impacts would both shock the materials and promote mixing between the impactor and target. This presumption seems to contradict the sample analysis results from Ryugu,

as mentioned earlier. Could the authors provide an explanation for this apparent discrepancy?

In line with the aforementioned concerns, I request the authors to provide specific details regarding the impact conditions encountered during the collisional evolutions. Can the authors elucidate on the impact velocity and the impactor-to-target size ratio as derived from their simulations? Following this, I would appreciate it if the authors could estimate the consequent increase in pressure and temperature resulting from the specified impact conditions. I hypothesize that if the impactor-to-target size ratio is close to unity, it would likely lead to enhanced material mixing. Could the authors comment on this?

To the first point - this work is meant to set the baseline expectation for collisional history, from which these other findings are interpreted. We have a more verbose response to a similar question to referee #1, and we did add a paragraph to the discussion of the paper describing the Nakamura findings and some implications - we think these explain this discrepancy.

To the second point, specific impact conditions are beyond the scope of this modeling effort for a few reasons that we try to describe here and with some more text in the manuscript. Primarily, the specific impact conditions cannot be extracted in any meaningful way because this work relies on probabilistic calculations of the collisional lifetime, which was derived in the works of Bottke et al. (2005) and Ballouz et al. (2020) that take into account the collisional probabilities and impact conditions etc. For us, in effect at each timestep we are rolling the dice to see if each asteroid has an impact that is as energetic as Q_D^* , if so, we break it up and move on. We are not modeling crossing orbits and tracking impact angles or impactor sizes and velocities. That being said, impact speeds are relatively uniform across the asteroid belt in the 5-6km/s range, impact angles fill the allowable distribution. The impactors required to reach Q_D^* level collisions are relatively small compared to the size of the target, and larger ratio impactors are increasingly unlikely.

The key paragraph where we discuss this in the Discussion, and we have added more text to better address these questions, is here;

“One of the simplifying assumptions of this model is that the only meaningful collisions are precisely at the energy where half of the mass is liberated from the target (Q_D^*). “, and we have added this to the end of that paragraph: **“While collisions substantially more energetic than Q_D^* are rare events, they could provide pathways to more widespread shock and mixing among the target and its reaccumulated remnants.** “

(3) In lines 185-190: The authors have posited that collisions with background asteroids primarily influence family members. However, immediately following the initial catastrophic collision that results in the formation of an asteroid family, the fragments would likely have orbits in close proximity to one another, given that they would share an almost identical point of impact. Consequently, right after the impact, there's a plausible scenario where these fragments (or family members) might collide with each other frequently, leading to further fragmentation. Could the authors provide a discussion or estimation on the collision timescale under such conditions?

Technically, our model can't account for this as it does not calculate impact probabilities - they are incorporated via the collisional lifetime calculation. Note that the initial velocity dispersion of the fragments, implemented at the request of Reviewer #2, finds that km-sized fragments can have initial orbits ~ 0.05 au away from a 100 km primary. So, while all fragments will briefly share an orbital node from the disruption event, they will have a large spread of orbital periods and then the action of Yarkovsky and orbital precession will drive them away really quickly from that shared node. This effect was discussed in Dell'Oro et al. (2002; Icarus). From their abstract:

"The results show the occurrence of a strong enhancement in the mutual collision rate among family members, immediately after family formation. Nevertheless, this episode lasts for a relatively short time, and it does not affect too severely the overall collisional evolution of the family. The early enhancement of mutual collisions, however, may influence the cratering record exhibited by the surfaces of family members, possibly laying the foundation for the early development of a surface regolith layer."

Meanwhile, Bottke et al. (2020; AJ) modeled of the crater size distribution on spacecraft observed asteroids, which included family members like Gaspra (with Flora) and Ida (with Koronis). They found nothing anomalous in the crater history of either one that would suggest a major contribution from family members. Until more evidence becomes available, we would argue it is reasonable to neglect the effect of intra-family member collisions on our test asteroids.

Minor comments:

Hirabayashi et al. 2020; Hyodo & Sugiura 2022 also discuss the shape change and resurfacing of small asteroids.

Yes, this is totally reasonable - we know YORP works out there in the Main Belt. We added this (I paste the whole paragraph for context):

“Similarly, YORP spinup and re-shaping has been attributed as the likely source of the distinct top-shape that both Bennu and Ryugu have and is common among NEAs and especially binary asteroids which account for 15% of the NEA population (Margot et al. 2015). YORP spinup is possibly a key or driving element of the spectral changes amongst S-type asteroids where movement of surface material is needed to liberate or expose fresh material that has not been exposed to space weather (see Graves et al. 2018). Therefore, **YORP spinup may provide a pathway to re-surface via landslides and re-shaping on timescales shorter than collisional lifetimes (Walsh et al. 2008, Harris et al. 2009, Scheeres 2015, Hirabayashi et al. 2020, Hyodo and Sugiura 2022)**. The YORP effect may also combine with the effect of tides during close flybys of the terrestrial planets or the effects of seismic shaking due to small impacts, to further contribute to the erasure of some cratering history on small asteroids (Richardson et al. 1998, Richardson et al. 2020, Jawin et al. 2020;2022).”

REVIEWERS' COMMENTS

Reviewer #1 (Remarks to the Author):

I believe the authors have appropriately addressed my concerns and I have no further comments.

Reviewer #2 (Remarks to the Author):

I've read Walsh et al.'s revised version of the manuscript "Bennu and Ryugu likely 2nd or later generation rubble piles." I appreciate the work the authors did in response to my comments. They adequately addressed the concerns raised in the first report and substantially improved the manuscript. Therefore, I recommend it for publication in Nature Communications.

Reviewer #3 (Remarks to the Author):

The authors have answered all my questions and concerns. I think the manuscript is now worthy of publication in Nature Communication.